# Decorin Secreted by Human Umbilical Cord Blood-Derived Mesenchymal Stem Cells Induces Macrophage Polarization via CD44 to Repair Hyperoxic Lung Injury

**DOI:** 10.3390/ijms20194815

**Published:** 2019-09-27

**Authors:** Ji Hye Kwon, Miyeon Kim, Yun Kyung Bae, Gee-Hye Kim, Soo Jin Choi, Wonil Oh, Soyoun Um, Hye Jin Jin

**Affiliations:** Biomedical Research Institute, MEDIPOST Co., Ltd., Seongnam 13494, Korea; kjh0127@medi-post.co.kr (J.H.K.); eldjfls3@medi-post.co.kr (M.K.); byk819@medi-post.co.kr (Y.K.B.); haha38@medi-post.co.kr (G.-H.K.); sjchoi@medi-post.co.kr (S.J.C.); wioh@medi-post.co.kr (W.O.)

**Keywords:** bronchopulmonary dysplasia, mesenchymal stem cell, macrophage polarization, decorin, CD44

## Abstract

Bronchopulmonary dysplasia (BPD), caused by hyperoxia in newborns and infants, results in lung damage and abnormal pulmonary function. However, the current treatments for BPD are steroidal and pharmacological therapies, which cause neurodevelopmental impairment. Treatment with umbilical cord blood-derived mesenchymal stem cells (UCB-MSCs) is an efficient alternative approach. To prevent pulmonary inflammation in BPD, this study investigated the hypothesis that a key regulator was secreted by MSCs to polarize inflammatory macrophages into anti-inflammatory macrophages at inflammation sites. Lipopolysaccharide-induced macrophages co-cultured with MSCs secreted low levels of the inflammatory cytokines, IL-8 and IL-6, but high levels of the anti-inflammatory cytokine, IL-10. Silencing decorin in MSCs suppressed the expression of CD44, which mediates anti-inflammatory activity in macrophages. The effects of MSCs were examined in a rat model of hyperoxic lung damage. Macrophage polarization differed depending on the levels of decorin secreted by MSCs. Moreover, intratracheal injection of decorin-silenced MSCs or MSCs secreting low levels of decorin confirmed impaired alveolarization of damaged lung tissues by down-regulation of decorin. In tissues, a decrease in the anti-inflammatory macrophage marker, CD163, was observed via CD44. Thus, we identified decorin as a key paracrine factor, inducing macrophage polarization via CD44, a master immunoregulator in mesenchymal stem cells.

## 1. Introduction

Bronchopulmonary dysplasia (BPD), a chronic lung disease in premature infants, results from supplemental oxygen and mechanical ventilation treatment, which can cause barotrauma, volutrauma, and oxygen toxicity [1,2,3]. Infants with BPD exhibit pathophysiological characteristics such as alveolar simplification, arrested lung growth, impaired vascular development, and abnormal pulmonary function [2,3,4,5]. To prevent pulmonary inflammation caused by mechanical ventilators, systemic corticosteroid treatment has been used and shown to reduce mortality [6,7]. There have been several reports regarding neurodevelopmental impairment in BPD patients with hydrocortisone regimens [8,9,10]. In addition to corticosteroids, pharmacological therapies, such as caffeine, diuretics, bronchodilators, and vitamin A, have been used for the prevention and management of BPD. However, it remains unclear how steroids and other medications reduce BPD, due to uncertainties regarding the dosage, timing, and choice of medication.

Stem cell therapy is a promising treatment for the regeneration of damaged lungs. With the common characteristics of self-renewal, clonogenic potential, and multipotency, mesenchymal stem cells (MSCs) are much more likely to be used in clinical applications [11,12,13]. Several experimental trials and clinical applications of MSCs have been attempted to treat incurable diseases [14]. There are several ongoing clinical trials of MSCs for the treatment and prevention of BPD [15,16,17]. Paracrine factors secreted by MSCs have the ability to suppress the immune response and regulate various immune cell functions [18,19]. The immunomodulatory effects of MSCs are communicated via MSC-secreted anti-inflammatory cytokines. However, the precise underlying mechanisms of MSC-mediated immunomodulation have not been fully clarified due to variations in the local microenvironment. Currently, the heterogeneity and variation in donor MSCs are the limitations of stem cell therapy. Therefore, these limitations in MSC immunomodulatory effects can be overcome by determining the appropriate criteria for highly efficient stem cells.

Recently, the treatment of BPD with MSCs has shown that hyperoxia-induced lung injury is ameliorated with alveolar and vascular remodeling [2,20,21]. Moreover, there is no evidence for the replacement of damaged lung tissue by engrafted MSCs. Secretome analysis has shown that the paracrine actions of MSCs are effective in BPD treatment [22,23]. Macrophages, derived from blood monocytes, reside in areas of tissue deterioration. Monocytes differentiate into classically activated macrophages (M1) when cells are exposed to microbicidal activity and they have an antigen-presenting function in tissue. Alternatively activated macrophages (M2) are associated with anti-inflammatory and homeostatic functions linked to fibrosis and wound healing. This diverges from the pro-inflammatory and antigen-presenting functions of M1, which induce the inflammatory cytokines, interleukin (IL)-1α, IL-6, IL-8, interferon (IFN)-γ, and tumor necrosis factor (TNF)-α. In contrast to M1, M2 induce the anti-inflammatory cytokines, IL-4, IL-10, IL-13, and Arg1 [24,25,26]. Paracrine factors secreted by MSCs facilitate the polarization from inflammatory M1 to anti-inflammatory M2 [27]. The broad range of potentially therapeutic proteins secreted by MSCs includes angiogenic factors, growth and trophic factors, chemokines, and anti-inflammatory cytokines [28,29]. In the inflammatory environment, TNF-inducible gene 6 (TSG-6) or prostaglandin E2 (PGE2), which are related to macrophage polarization, are released by MSCs [30,31,32]. However, these factors are insufficient to explain all of the anti-inflammatory effects and signaling mechanisms of MSCs. Additional proteins are needed to explain the anti-inflammatory reactions and to control macrophage polarization.

Decorin is a small secreted leucin-rich proteoglycan that influences proteins involved in apoptosis, cell proliferation, transcription, chemotherapy resistance, mitosis, and fatty acid metabolism [33,34,35,36]. Recent research also suggests that decorin-modified MSCs attenuate lung injury through anti-inflammatory and anti-fibrotic activities. Decorin-overexpressing umbilical cord blood-derived mesenchymal stem cells (UCB-MSCs) have been shown to evoke a reduction in up-regulation of the following chemokines and inflammatory cytokines: MCP-1, MCP-3, MIP-2, eotaxin, IFN-γ, IL-12, and TNF-α [37]. CD44 is relatively upregulated in macrophages recruited following tissue injury and is implicated in many chronic inflammatory diseases [38]. In this study, we determined the criteria for highly efficient stem cells based on their anti-inflammatory effects. After pre-screening the proteins secreted by MSCs, decorin was selected and confirmed as a potential marker. To verify the therapeutic effect of MSCs on BPD, macrophage polarization involving interactions between decorin and CD44 was investigated. We found that the key regulator, decorin, and the underlying mechanisms of macrophage polarization, enhanced the anti-inflammatory effects of MSCs.

## 2. Results

### 2.1. MSCs Induced Macrophage Polarization

To investigate macrophage polarization triggered by MSCs and to stimulate paracrine actions for clinical application, we analyzed rat alveolar macrophages (NR8383) stimulated with lipopolysaccharide (LPS) and co-cultured with 1.9 × 10^4^ MSCs for 3 days in a direct cell-to-cell contact culture system. Immunofluorescence staining was used to identify classically activated macrophages (inflammatory macrophages, M1) and alternatively activated macrophages (anti-inflammatory macrophages, M2) with antibodies against the M1 marker, CD11b (red) and the M2 marker, CD163 (green). Macrophages induced by LPS showed increased expression of CD11b. However, LPS-induced macrophages co-cultured with MSCs for 3 days showed a reduction in CD11b expression (Figure 1a,a’). Conversely, CD163 expression was significantly increased in macrophages co-cultured with MSCs compared to LPS-activated MSCs (Figure 1b,b’). Our data suggested that MSCs polarize M1 into M2.

Furthermore, cytokines produced by macrophages and/or MSCs were measured by enzyme-linked immunosorbent assay (ELISA). Supernatants of cultured macrophages showed increased levels of the pro-inflammatory cytokines, IL-8 and IL-6, after LPS induction. By contrast, levels of the anti-inflammatory cytokine, IL-10, decreased on LPS-treated macrophages. However, MSCs suppressed M1 activation, resulting in a decrease in IL-8 and IL-6 expression and an increase in IL-10 expression. Three different MSCs were prepared and tested to determine the ability of MSCs to induce macrophage polarization (Figure 1c–e). These results were consistent with immunofluorescence results, which showed that MSCs trigger macrophage polarization.

### 2.2. MSC-Related Macrophage Polarization Was Associated with CD44 on Macrophages

To identify the main factor responsible for the change in macrophage functional polarity from M1 toward the anti-inflammatory M2 phenotype, we focused on CD44. It has been reported that CD44 regulates macrophage recruitment to the lungs in inflammatory airway diseases [38]. Moreover, CD44, mediated by TSG-6, activates the macrophage anti-inflammatory program [39].

We silenced CD44 in macrophages co-cultured with MSCs to examine the involvement of CD44 on macrophage polarization. To determine whether CD44-dependent macrophage polarization is related to the action of MSCs, we transfected LPS-stimulated macrophages with CD44 siRNA or a scrambled control siRNA. Within 24 h of incubation with 50 nM CD44 siRNA followed by LPS stimulation, CD44 protein expression on LPS-stimulated macrophages was clearly suppressed (Figure 2a). Immunofluorescent staining results showed that LPS-stimulated macrophages had a 2.5-fold increase in CD11b expression, but a three-fold decrease in CD163 expression compared to inactivated macrophages (Figure 2b,b’,c,c’). Interestingly, CD44 expression on macrophages decreased by approximately 20% after LPS stimulation (Figure 2d,d’). However, MSCs suppressed CD11b expression on LPS-stimulated macrophages. Significant increases in CD163 and CD44 expression were also noted on macrophages co-cultured with MSCs. After 3 days of co-culturing MSCs with CD44-siRNA-transfected macrophages, there were slight increases of CD11b expression, but significant decreases in CD163 expression. These results demonstrated that CD44 is a key modulator of macrophage polarization and is regulated by MSCs (Figure 2b,b’–d,d’). Based on the cytokine levels secreted by macrophages, CD44 silencing in macrophages tended to restrict macrophage polarization triggered by co-culture with MSCs. ELISA results demonstrated an increase in the production of the inflammatory cytokines, IL-8 and TNF-α, and a significant decrease in the production of the anti-inflammatory cytokine, IL-10, on CD44-siRNA-transfected macrophages compared with control macrophages after co-culture with MSCs (Figure 2e–g). Overall, our data suggested that CD44 expression on macrophages, which is related to macrophage polarization, was affected and triggered by MSCs.

### 2.3. Decorin Secreted by MSCs Was a Key Modulator of Macrophage Polarization

We pre-screened the secretome of human MSCs to select candidate factors affecting immunomodulation. Protein expression levels on MSCs co-cultured with macrophages were measured using a fluorescent human antibody array normalized to MSCs only. The therapeutic effects of MSCs were assessed by the anti-inflammatory factors that they secreted. Levels of the anti-inflammatory paracrine factors, VEGF, PTX3, TIMP-2, TSP-1, and decorin, secreted by MSCs co-cultured with macrophages, were determined in the secretome analysis (Appendix A). Previous reports have suggested that decorin-overexpressing UCB-MSCs have anti-inflammatory effects [37]. Among selective paracrine factors secreted by MSCs co-cultured with macrophages, the intensity of decorin showed a 15-fold change compared to MSCs only in secretome analysis data.

To test the role of MSC-derived decorin on inflammatory conditions, we suppressed the secretion of decorin by transfecting MSCs with decorin siRNAs for 24 h (Appendix A). Approximately 100 nM scrambled control siRNA or decorin siRNA were used to study the effects of decorin suppression. LPS-stimulated macrophages co-cultured with MSCs or scrambled siRNA-transfected MSCs (Con siR-MSCs) showed an increase in CD44 expression (Figure 3c,c’), whereas decorin siRNA-transfected MSCs (Decorin siR-MSCs) showed an increase in the expression of the inflammatory macrophage marker, CD11b (Figure 3a,a’), and a decrease in the expression of the anti-inflammatory macrophage marker, CD163 (Figure 3b,b’). The macrophage polarization effects of MSCs were abolished by transfection with decorin siRNA.

Expression of the inflammatory cytokines, IL-8 and IL-6, which are secreted by rat macrophages, was found to increase on macrophages co-cultured with decorin siRNA-transfected MSCs. In contrast, the levels of the anti-inflammatory cytokine, IL-10, were significantly suppressed on macrophages co-cultured with decorin siRNA-transfected MSCs compared with MSCs only and macrophages co-cultured with scrambled control siRNA-transfected MSCs (Figure 3d–f). These results demonstrated that the paracrine factor, decorin, secreted by MSCs, is involved in polarizing inflammatory macrophages into anti-inflammatory macrophages. CD44 present on the macrophages, which was regulated by decorin from MSCs, was a key modulator controlling macrophage polarization.

### 2.4. Treatment with MSCs Reduced Local Inflammatory Responses in a Rat Hyperoxic Lung Injury Model

Decorin levels on MSCs under inflammatory conditions were measured in 10 different MSC lots to verify the effects of decorin on MSCs. Based on the level of decorin, MSCs from 10 different donors were divided into two groups: MSC-high (H) and MSC-low (L). There was a significant difference in decorin levels between MSC-H and MSC-L groups. The concentrations of decorin after co-culture with LPS-activated macrophages for 3 days were determined by ELISA. The results showed varying levels of decorin (Figure 4a).

We hypothesized that the level of decorin secreted by MSCs in an inflammatory environment would determine the control of anti-inflammatory reactions by MSCs. We next examined whether MSCs caused immunomodulation by polarizing macrophages into anti-inflammatory macrophages in rats with hyperoxia-induced lung injury. To create a model of BPD, rat pups were raised under hyperoxic conditions, with 90% oxygen, for 14 days. The experimental scheme is described in Appendix A. Of the MSC lots, lot MSC H was selected as an MSC-H group and lot MSC L was selected as an MSC-L group. MSCs were injected intratracheally at postnatal day 5 (P5) to compare the therapeutic effects of the different MSC types. The MSC H lot had the lowest decorin levels of the MSC-H group, whereas the MSC L lot had the highest decorin levels of the MSC-L group. We injected cells from lots MSC H and MSC L into rats with severe hyperoxic lung injury and then sacrificed them on P14. In the normoxic control group had 100% of survival rate. At P14, only approximately 20% of hyperoxic lung injured (BPD) rats had survived. However, MSC H treatment increased the survival rate to 80% and MSC L treatment increased the survival rate to 75%. With MSCs treatment on BPD, the survival rates increased 20% into 75% and 80%. These results demonstrated that MSCs increased the probability of survival by inducing the recovery from lung damage (Figure 4b). Histological and morphometrical analyses were performed on rat lung tissues using representative optical microscopy photomicrographs. The degree of alveolarization was then assessed using the mean linear intercept (MLI). MLI was significantly lower in BPD rats after MSC H treatment. MSC L treatment also slightly decreased MLI, which indicated the recovery of impaired alveoli (Figure 4c). Hematoxylin and eosin staining identified several types of inflammatory cells infiltrating the interstitium of the lung and fewer and larger simplified alveoli in rats with BPD. Variable interstitial fibroproliferation and fewer and dysmorphic capillaries were also observed in the lungs of rats with hyperoxia-induced BPD. There was no significant change in symptoms in the BPD model after MSC L treatment. However, MSC H treatment increased the number of normal alveoli and decreased immune cell infiltration (Figure 4d). This indicated that cells from lot MSC H had a significantly greater effect than cells from lot MSC L. These results suggested that the level of decorin production determined the outcome when treating impaired alveolarization.

Rat lung tissues from the BPD model and the MSC treatment groups were immunostained for the inflammatory marker, CD11b, and the anti-inflammatory marker, CD163. CD11b was highly expressed in lung tissue from BPD rats, whereas CD163 expression was markedly suppressed. After the injection of cells from lot MSC H, BPD rat lungs clearly showed high levels of CD163 expression and low levels of CD11b expression. BPD rats injected with cells from lot MSC L also showed a significant difference in CD163 expression compared to untreated BPD rats (Figure 4e,e’,f,f’). Moreover, the number of CD44-positive macrophages increased in BPD rats after the injection of cells from both MSC H and MSC L lots (Figure 4g,g’). The transplanted human MSCs were confirmed by staining the human β2 microglobulin of human MSCs (Figure 4h,h’). Next, the pro-inflammatory cytokines, IL-8 and IL-6, and the anti-inflammatory cytokine, IL-10, were analyzed using in rat lung bronchoalveolar lavage fluid (BALF) at P14. Elevations in IL-8 and IL-6 levels were observed in BPD rat BALF, but MSC H treatment significantly decreased the levels of these cytokines. By contrast, IL-10 levels were increased by MSC H treatment. Interestingly, MSC L treatment also resulted in increased levels of IL-10, but had no effect on IL-8 or IL-6 levels. Overall, the difference in decorin levels between MSC H and MSC L resulted in different levels of repair of hyperoxic lung damage in rats (Figure 4i–k). Taken together, the levels of decorin secretion by MSCs determined the outcome of BPD treatment.

### 2.5. Decorin Regulated the Therapeutic Effects of MSCs by Recruiting CD44 on Macrophages

The effects of decorin on anti-inflammatory modulation were evaluated by the injection of decorin siRNA-transfected MSCs into BPD rats. The 20% survival rate of BPD rats increased to 70% after treatment with MSCs, to 80% after treatment with scrambled control siRNA-transfected MSCs, and to 40% after treatment with decorin siRNA-transfected MSCs (Figure 5a). Treatment with naïve MSCs or with scrambled control siRNA-transfected MSCs resulted in the reconstruction of alveoli in damaged lung tissue. The MLI values for MSC-treated and scrambled control siRNA-transfected MSC-treated BPD rats did not differ from the MLI values in the normal group. However, the decorin siRNA-transfected MSC-injected BPD rats demonstrated similar results to the BPD control group without MSC treatment. Abnormal alveoli were observed in BPD and decorin-knockdown MSC-injected BPD groups. MLI levels were also higher in the decorin-knockdown MSC-injected BPD group compared with the normal groups (Figure 5b,c). The high levels of CD11b expression in the BPD group were suppressed by MSC treatment, but lower levels of CD163 expression were observed in BPD rats. However, decorin-knockdown MSC-treated BPD rats showed an increase in CD11b expression, but a decrease in CD163 expression of BPD rats and decorin-knockdown MSC-injected BPD rats had lower levels of CD44 on macrophages (Figure 5d,d’–f,f’). The presence of injected MSCs was confirmed by staining for hβ2MG (Figure 5g,g’). An analysis of BALF samples was performed using ELISA, to compare the secretion of IL-8, IL-6, and IL-10 in the different groups. An increase in the levels of the pro-inflammatory cytokines, IL-8 and IL-6, and a decrease in the levels of the anti-inflammatory cytokine, IL-10, were observed in BPD rats. The same results were observed in BALF samples from decorin-knockdown MSC-injected BPD rats. However, BPD rats treated with MSCs or with scrambled control siRNA-transfected MSCs showed decreased levels of IL-8 and IL-6, but increased levels of IL-10 (Figure 5h,j). These data cooperated with in vitro results from decorin-knockdown MSCs co-cultured with LPS-stimulated macrophages (Figure 3). Both in vitro and in vivo experiments indicated that the paracrine actions of decorin are responsible for macrophage polarization, via an increase in CD44 expression.

## 3. Discussion

In this study, we investigated the therapeutic effects of UCB-MSCs on BPD and the paracrine factors involved in macrophage polarization. CD44 knockdown in NR8383 rat alveolar macrophages reduced macrophage polarization when co-cultured with MSCs. The transition between M1 and M2 is caused by paracrine actions of MSCs and is associated with therapeutic effects. The paracrine actions involved in macrophage polarization were also abolished in vivo in a newborn rat model of hyperoxic lung injury by siRNA-mediated knockdown of decorin, but not by a scrambled control siRNA. Taken together, our results demonstrated that decorin secreted by MSCs is a key modulator of macrophage polarization to regulate anti-inflammatory reactions.

BPD is a severe chronic lung disease in newborns and infants, caused by hyperoxic conditions due to the long-term use of mechanical ventilation for oxygen supplementation. It results in long-term breathing difficulties and significant morbidity. The rationale for the prevention and treatment of BPD is unclear [4,40]. Recently, preclinical studies have been performed to assess MSC injection for the treatment of hyperoxia-induced animal models of BPD [41,42,43,44,45]. The therapeutic effects of UCB-MSCs in BPD have been confirmed in ongoing clinical trials worldwide [15,16,46]. Previous studies have demonstrated that immunomodulation by MSCs also regulates the functional of immune cells, including monocytes/macrophages, T cells, B cells, and natural killer cells [47,48,49,50,51,52]. Our results showed that MSCs direct the immunological fate of macrophages. LPS-activated macrophages directly co-cultured with MSCs polarized from an M1 phenotype to an M2 phenotype. Anti-inflammatory M2 expressed high levels of CD163, but low levels of the inflammatory marker, CD11b. Secretion of the anti-inflammatory cytokine, IL-10, by rat macrophages increased after co-culture with MSCs. By contrast, the levels of the pro-inflammatory cytokines, IL-8 and IL-6, significantly decreased in LPS-stimulated macrophages after co-culture with MSCs. We tested three MSC lots from different donors to determine the variation in the immunomodulation capacity of MSCs. These results indicated that MSCs drive the polarization of macrophages towards a less inflammatory state and simultaneously enhance the recovery of damaged tissues.

Next, the cognate receptor on macrophages by which MSCs trigger polarization was investigated. We focused our attention on CD44, a surface glycoprotein known as a controller of the macrophage fusion. According to previous studies, CD44 plays an important role in pulmonary innate immunity. Decreased macrophage recruitment and decreased expression of anti-inflammatory cytokines are observed at the site of lung injury in CD44-knockout mice [38]. CD44 is a principal receptor for hyaluronan (HA) to activate leukocytes and parenchymal cells at sites of inflammation. Moreover, TSG-6, an anti-inflammatory cytokine mainly secreted by MSCs, enhances the interaction of CD44 with HA in inflammatory environments. CD44 triggers fibroblast migration via TGFβ activation, to repair damaged tissue [39,53]. In our study, the beneficial effects of MSCs observed in vitro were abolished by CD44 knockdown in macrophages co-cultured with MSCs. CD44-silenced macrophages failed to polarize into the M2 phenotype after MSC treatment. Levels of the pro-inflammatory cytokines, IL-8 and IL-6, in CD44 knockdown macrophages co-cultured with MSCs did not change compared to their levels in LPS-activated macrophages. Thus, the depletion of CD44 may affect macrophage polarization induced by MSCs. Taken together, these results indicate that CD44 on macrophages is a key modulator of MSC-induced polarization.

Decorin is a small leucine-rich proteoglycan that is closely related in structure to biglycan protein, which induces a Toll-like receptor (TLR) 2/4-dependent signaling cascade in macrophages [54,55]. M1 recruitment into the kidney has been studied in renal ischemia/reperfusion injury (IRI) in CD44 knockout mice. In this study, the biglycan-CD44 interaction via TLR4 enhanced M1 autophagy, resulting in an increase in the number of M2, followed by a reduction in tubular damage in the kidney. Previous reports have also suggested that decorin induces Peg3, a master regulator of M1 autophagy [56]. As a co-receptor for biglycan, CD44 is a key regulator of macrophage recruitment [57]. The role of decorin on macrophages has also been reported related to macrophage proliferation through induction of p27^Kip1^ and p21^Waf1^ [58]. There have been previous reports regarding the therapeutic effects of decorin. Over-expression of decorin on UCB-MSCs also attenuates acute inflammation after radiation-induced lung injury in mice by regulating inflammation and immune responses [37]. Moreover, decorin treatment of scarring that occurs after penetrant central nervous system injury, results in regeneration of axons in the spinal cord [59]. Overexpression of decorin ameliorates diabetic cardiomyopathy by promoting angiogenesis [60]. Furthermore, bone marrow-derived mesenchymal stem cells infected with decorin-expressing adenovirus promote the recovery of liver function after fibrotic injury [61]. Following the result of secretome analysis (Appendix A), the major increase of VEGF, PTX3, TIMP-2, TSP-1, and decorin was demonstrated. The slight increase of TGF-β1 and decrease of EGF were observed. There may be other critical secreted factors that act similar to decorin, and our research did not preclude this possibility.

Previous and ongoing studies support our pre-screening analysis (Appendix A), which investigated secreted proteins in co-cultures of MSCs with LPS-activated macrophages and found significant increases in decorin levels. Inflammatory environments induce the production of paracrine factors by MSCs to regulate immune cells. The paracrine factors secreted by MSCs are important to screen to enable the identification of highly efficient MSCs for therapeutic applications. In our study, MSCs were isolated from 10 different donors and co-cultured with LPS-activated macrophages to construct inflammatory environments. The levels of decorin varied depending on the MSC lot co-cultured with LPS-activated macrophages. MSCs were divided into two groups: Those expressing high levels of decorin (MSC-H) and those expressing low levels of decorin (MSC-L). Decorin levels on MSCs determine the extent of recovery of damaged lung tissues in BPD. In addition, we silenced decorin in MSCs to confirm its role in MSC-induced macrophage polarization. MSCs and scrambled siRNA-transfected MSCs polarized macrophages into the M2 phenotype, with high levels of CD163 expression and low levels of CD11b expression. CD44 expression levels on macrophages were up-regulated by MSC treatment. However, this effect was no longer present when MSCs were pretreated with decorin siRNA. Decorin silencing suppressed CD44 expression in MSCs. The anti-inflammatory effects of decorin secreted by MSCs were indicated by a decrease in the secretion of the pro-inflammatory cytokines, IL-8 and IL-6, and enhanced secretion of IL-10 by co-cultured macrophages. Macrophages co-cultured with decorin-silenced MSCs showed an increase in IL-10 secretion and a decrease in IL-8 and IL-6 secretion, compared with macrophages co-cultured with MSCs or scrambled siRNA-transfected MSCs. Thus, our results confirmed that decorin secreted by MSCs attenuated inflammation via interaction with CD44 on macrophages.

In a previous study, we showed that MSCs transform the phenotype and functional properties of macrophages in BPD [19,41]. In a BPD model, MSCs were shown to trigger the recovery of injured lung tissues. To confirm decorin as a candidate paracrine factor for the selection of the efficient MSCs, we intratracheally injected MSCs with different levels of decorin expression. The MSC H lot secreted the lowest levels of decorin in the high-expressing group, whereas the MSC L lot secreted the highest levels of decorin in the low-expressing group. BPD rats injected with cells from the MSC H lot showed a recovery of lung tissue, with normal alveoli. MLI levels in BPD rats treated with cells from lot MSC L were not significantly different from those in BPD control rats. Interestingly, there was no difference in survival rates between BPD rats injected with cells from lot MSC H and those injected with cells from lot MSC L. However, a significant decrease in CD11b levels and an increase in CD163 levels were observed in BPD rats injected with cells from lot MSC H. In contrast, BPD rats injected with cells from lot MSC L, which secreted low levels of decorin, had high levels of CD11b expression and low levels of CD163 expression. CD44 expression levels on macrophages were high in BPD rats injected with cells from lot MSC H. The anti-inflammatory effect of lot MSC H was confirmed by both a decrease in the levels of the pro-inflammatory cytokines, IL-8 and IL-6, and an increase in IL-10 levels in lung BALF. Although similar survival rates were seen between BPD rats injected with cells from the MSC H lot and those injected with cells from the MSC L lot, the therapeutic effects of MSC L were greater than MSC H, based on the triggering of macrophage polarization.

To confirm the critical role of decorin secreted by MSCs, decorin-silenced MSCs were injected into BPD rats. Injection of decorin-silenced MSCs did not result in full recovery of damaged lung tissues in BPD rats or the suppression of CD11b expression. Moreover, the expression of CD163 was not detected in BPD rats injected with decorin-silenced MSCs. The suppression of CD44 expression seen in BPD rats injected with decorin-silenced MSCs suggested that decorin is a key regulator of macrophage polarization via CD44. A major question that remains unanswered is whether CD44 induces the TLR4-dependent activation of M1 autophagy, leading to M2 macrophage polarization. Therefore, further research regarding M1 autophagy caused by decorin/CD44 interactions is required. Levels of the pro-inflammatory cytokines, IL-8 and IL-6, in lung BALF were increased in BPD rats injected with decorin-silenced MSCs compared to those injected with MSCs or scrambled siRNA-transfected MSCs. In contrast, IL-10 secretion was suppressed in BPD rats injected with decorin-silenced MSCs. Overall, our results suggested that decorin is a key regulator of macrophage polarization by triggering CD44.

In conclusion, the protective effects of MSC therapy against hyperoxia-induced lung injuries are mediated primarily by their anti-inflammatory effects rather than by their regenerating capacity. Thus, specifically triggering decorin-induced macrophage polarization by MSCs may be a novel strategy for the prevention and therapy of BPD. Decorin triggers CD44-dependent macrophage polarization and repair of damaged lung tissues in BPD. Overall, we postulate that the efficacy of stem cell-mediated therapy in inflammatory diseases and tissue damage could be determined by the paracrine factors secreted from MSCs. The anti-inflammatory effects of macrophage polarization triggered by decorin, via CD44, may contribute to the therapeutic efficacy of MSCs.

## 4. Materials and Methods

### 4.1. Cell Preparation and Culture Conditions

UCB samples were collected from human umbilical veins isolated within 24 h after neonatal delivery after informed maternal consent. This protocol was approved by the Institutional Review Board of MEDIPOST Co., Ltd. (MP-2015-6-4). UCB-MSCs, separated from mononuclear cells (MNCs) using Ficoll-Paque™ PLUS (GE Healthcare, Uppsala, Sweden), were washed and cultured in minimum essential medium alpha (α-MEM; Gibco/Invitrogen, Carlsbad, CA, USA) supplemented with 10% fetal bovine serum (FBS; Gibco, Thermo Fisher Scientific, Waltham, MA, USA) at 37 °C in a 5% CO_2_ incubator. The culture medium was changed once every 2–3 days, as previously described [62,63]. The basic characteristics of the MSCs, such as stemness and multi-lineage potential, are shown in Appendix A.

The UCB-MSCs used for these experiments were at passage 6 (p6). To clarify the characteristics of MSCs, we determined the proliferation using population doubling level (PD) calculation and senescence on passages 4 and 6. Each passage was cultured for 5 days and analyzed with the trypan blue exclusion method as described previously [64]. The population doubling level using the following formula: PD = log (total viable cells at harvest/total viable cells at seed)/log2. The senescence of MSCs was detected by senescence-associated β-galactosidase (β-Gal) staining (Sigma-Aldrich, St. Louis, MO, USA) followed by manufacturer’s instructions.

### 4.2. In Vitro Inflammation Conditions

Rat alveolar macrophage (NR8383 cells), were purchased form ATCC (American Type Culture Collection, Manassas, VA, USA) and were cultured in F-12K medium with 15% FBS. NR8383 cells (1 × 10^5^) were activated with 1 μg/mL LPS derived from *Escherichia coli* I55:B5 (Sigma-Aldrich, St. Louis, MO, USA). NR8383 cells activated with LPS were used as a positive control for inflammation. LPS-activated NR8383 cells were co-cultured with 1.9 × 10^4^ MSCs for 3 days. To simulate paracrine actions in a clinical application setting, a direct cell-to-cell contact culture system was used. The supernatants were collected from activated NR8383 cells cultured with MSCs. The quantitative measurement of rat IL-6, IL-8, IL-10, and TNFα (all from R&D systems, Minneapolis, MN, USA) and human decorin (Sigma-Aldrich) levels in supernatants were performed using ELISA.

### 4.3. Small Interfering RNA-Mediated Knockdown of Target Genes

Human decorin siRNAs (100 nM), rat CD44 siRNAs (50 nM), and scrambled siRNAs (50 and 100 nM; Dharmacon, Lafayette, CO, USA) were transfected for 24 h using DharmaFECT reagent, as recommended by the manufacturer. The sequences of primers used for target genes are described listed in Table 1.

### 4.4. Immunofluorescent Staining

NR8383 cells were incubated with rat monoclonal primary antibody (CD11b, 1:100; Abcam, Cambridge, UK) followed by the appropriate secondary antibody (Cy3-conjugated secondary antibody, 1:350; Jackson ImmunoResearch Europe Ltd., Newmarket, UK). Cell surface glycoproteins CD163 (1:50; Santa Cruz Biotechnology, Dallas, TX, USA), CD44 (1:150; Novus Biologicals, Centennial, CO, USA) on fixed NR8383 cells were stained with rat monoclonal antibodies, followed by Alexa Fluor^®^ 488 (1:350, Jackson ImmunoResearch). Injected human MSCs were detected with an anti-human β2 microglobulin (hβ2MG) antibody (1:100, Santa Cruz, Dallas, TX, USA) with an Alexa Fluor^®^ 488-conjugated secondary antibody (1:350, Jackson ImmunoResearch). Before co-culturing with human MSCs, NR8383 cell nuclei were counterstained with Hoechst 33342 to prevent the staining of human nuclei. Fluorescent images were acquired and analyzed using an LSM 800 confocal microscope (Zeiss, Oberkochen, Germany).

### 4.5. Western Blotting

Macrophages were transfected with a scrambled siRNA or a rat CD44 siRNA for 24 h. Transfected macrophages were lysed with RIPA buffer to extract protein. A total of 20 μg of each protein extract was electrophoresed on a sodium dodecyl sulfate-polyacrylamide (SDS-PAGE) gel and then transferred to a nitrocellulose membrane. Blocked membranes were incubated with a primary anti-CD44 antibody (Novus Biologicals, Centennial, CO, USA), followed by horseradish peroxidase-conjugated secondary antibodies. Chemiluminescent intensity of immunoblotted bands was visualized using a ChemiDoc Imaging System (Bio-Rad, Hercules, CA, USA). The intensity of each band was normalized to β-actin band intensity (Novus Biologicals).

### 4.6. Flow Cytometry

The characteristics of MSCs (p6) isolated from ten independent donors were analyzed by flow cytometry. MSCs were collected and labeled with fluorescein isothiocyanate (FITC)-conjugated human CD14, CD45, and human leukocyte antigen (HLA)-DR antibodies (BD Biosciences, Franklin Lakes, NJ, USA). Phycoerythrin (PE)-conjugated human CD73, CD166 (BD Biosciences), CD90, and CD105 (Invitrogen, Carlsbad, CA, USA) antibodies were used to measure stem cell surface markers on MSCs. An isotype control was also included. Washed MSCs were fixed with 1% (v/v) paraformaldehyde (Sigma-Aldrich). Stained cells were analyzed by flow cytometry on a MACSQuant instrument (Miltenvi Biotec, Bergisch Gladbach, Germany).

### 4.7. Cell Differentiation

The multi-lineage differentiation capacity of MSCs (p4 and p6) was determined by analyzing selective differentiation marker expression. The conditions used for osteogenic, chondrogenic, and adipogenic differentiation of MSCs were adopted from previous studies [65]. To evaluate osteogenesis, cells differentiated into osteoblasts or osteocytes were stained with alkaline phosphatase (Stemgent, Cambridge, MA, USA). Chondrogenic differentiation was assessed by Safranin O staining (Sigma-Aldrich), which detects proteoglycans on cartilage-like cells obtained from pellet culture. Oil red O staining (Sigma-Aldrich) was performed to detect accumulated lipid droplets in differentiated cells.

### 4.8. Animal Model

All animal experiments were reviewed and approved by the Institutional Animal Care and Use Committee of MEDIPOST Co., Ltd. (MP-LAR-2017-1-2). This study was also performed in accordance with the institutional and National Institutes of Health guidelines for laboratory animal care. Rat pups (10 g) were delivered from timed pregnant Sprague-Dawley rats (Samtako Bio Korea Co. Ltd., Osan, Korea). Two experimental designs were used, as described in Table 2.

Within 10 h after birth, rat pups were randomly assigned to following four groups for the first in vivo experiment: Normoxic control group (normal), hyperoxic lung injury group (BPD), hyperoxic lung injury group with MSC H (BPD + MSC H), or MSC L (BPD + MSC L). The differences between the MSC H group and MSC L group were determined by measuring secreted decorin levels. The second experimental design had five groups: (i) Normal, (ii) BPD, and BPD with (iii) MSCs (BPD + MSC), (iv) scrambled siRNA-treated MSCs (BPD + Con siR-MSC), or (v) decorin siRNA-treated MSCs (BPD + Decorin siR-MSC). The control group was maintained under normoxic conditions, whereas the hyperoxic groups were exposed to hyperoxic chambers in which 90% oxygen was maintained from birth to postnatal (P) day 14, as reported previously [41]. To avoid oxygen toxicity, nursing mother rats were rotated daily between litters maintained under normoxic and hyperoxic conditions. MSCs (1 × 10^5^, p6) were washed with saline after washing twice with pre-warmed MEM-α without phenol red. After saline washing, MSC suspensions were prepared in saline and injected intratracheally at P5, as described previously [42]. The survival rates and health conditions of all rat pups were monitored daily. Rat pups were anesthetized by an intraperitoneal injection of pentobarbital (60 mg/kg) at P14. Eleven to fifteen animals were assigned per group.

### 4.9. In Vivo Transplantation Immunohistochemistry and Morphometry

Whole lung tissues were obtained from sacrificed rat pups and were fixed with 4% paraformaldehyde. Fixed lung tissues were embedded in paraffin, sectioned, and then stained with hematoxylin and eosin (H&E). MLI was used to measure the level of alveolarization by dividing the total length of lines drawn across the lung section by the number of intercepts encountered, as reported previously [42]. Randomly selected sections (>3) per rat and 100 fields per section were assessed. To confirm the transplantation of injected MSCs in lung tissues, an anti-hβ2MG antibody was used and visualized with an Alexa Fluor^®^ 488-conjugated secondary antibody. To detect rat alveolar macrophages, CD11b, CD163, and CD44 were stained with primary antibodies, followed by detection with Alexa Fluor^®^ 488- or Cy3-conjugated secondary antibodies. Nuclei in lung tissues were counterstained with Hoechst 33342. Stained lung tissues were imaged and analyzed by LSM 800 confocal microscopy. The concentrations of rat IL-8, IL-6, and IL-10 in the BALF samples were determined using ELISA, as described previously [41,42].

### 4.10. Statistical Analysis

All data are presented as means ± standard deviations (SDs) of the values obtained in experiments performed at least in triplicate. Statistical analysis was performed using a one-way analysis of variance, followed by a least-significant difference (LSD) post-hoc test with Prism 6 software (GraphPad, San Diego, CA, USA). A statistically significant difference was reported if *p* < 0.05.

## Figures and Tables

**Figure 1 ijms-20-04815-f001:**
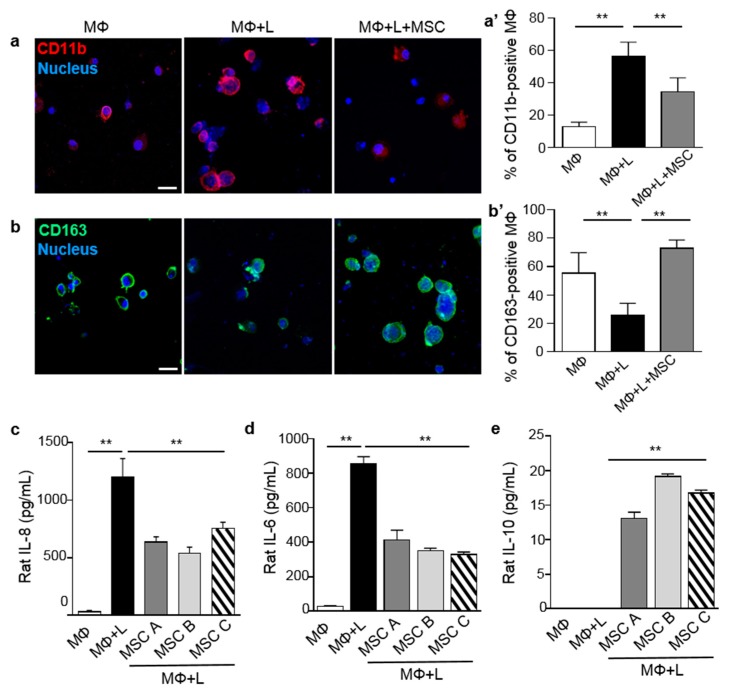
Mesenchymal stem cells (MSCs) attenuate M1 activation and facilitate M2 activation. (**a**,**a’**) Rat alveolar macrophages (NR8383) stimulated by lipopolysaccharide (LPS) were co-cultured with MSCs for 3 days. LPS-activated macrophages expressed the M1 marker, CD11b. (**b**,**b’**) The number of macrophages stained for the M2 marker, CD163, increased after co-culture with MSCs. CD11b-positive macrophages and CD163-positive macrophages were analyzed according to the percentage of CD11b- and CD163-positive cells. Red (CD11b) and green (CD163) staining indicated cells positive for these markers. Nuclei were stained with Hoechst 33342. (**c**–**e**) The secretion of the inflammatory markers, IL-8 and IL-6 and the anti-inflammatory marker, IL-10, by macrophages were measured by ELISA. Three different MSCs were tested to confirm the effects of MSCs. All MSCs predisposed macrophages to polarization and subsequent secretion of the anti-inflammatory cytokines indicative of the M2 phenotype. Scale bar = 20 μm. Data are presented as mean ± SD, *n* = 5 per group. ** *p* < 0.01. MΦ, macrophage; L, LPS.

**Figure 2 ijms-20-04815-f002:**
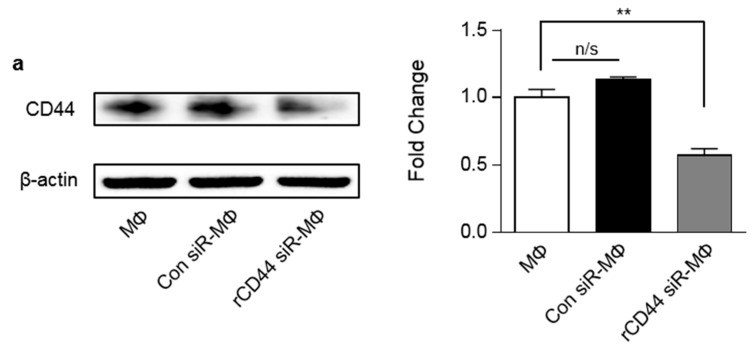
Modulation of macrophage functional polarity towards an anti-inflammatory phenotype by MSCs occurs via CD44 on rat macrophages. (**a**) Rat macrophages were transfected with scrambled siRNA (50 nM) or rat CD44 siRNA (50 nM) for 24 h. After CD44 siRNA transfections, LPS stimulation was performed. Rat CD44 protein expression on macrophages was analyzed by western blotting. Intensity was normalized to macrophages only. (**b**,**b’**–**d**,**d’**) CD44 knockdown in macrophages disrupted macrophage polarization. Confocal microscopy of CD44 siRNA-transfected macrophages showed an increased expression of the M1 marker, CD11b and decreased expression of the M2 marker, CD163, despite treatment with MSCs. Quantitative analysis showed the significant results of knockdown CD44 on macrophages. Nuclei were stained with Hoechst 33342. Red (CD11b) and green (CD163 and CD44) staining indicate positive cells. (**e**–**g**) Cytokine analysis was performed by ELISA. Supernatants from CD44 siRNA-transfected macrophage cultures also showed high levels of IL-8 and TNF-α, but low levels of IL-10. Scale bar = 20 μm. Data are presented as mean ± SD, *n* = 5 per group. ** *p* < 0.01, * *p* < 0.05. n/s, not significant; MΦ, macrophage; L, LPS; Con siR, scrambled siRNA-transfected control group; rCD44 siR, rat CD44 siRNA-transfected group.

**Figure 3 ijms-20-04815-f003:**
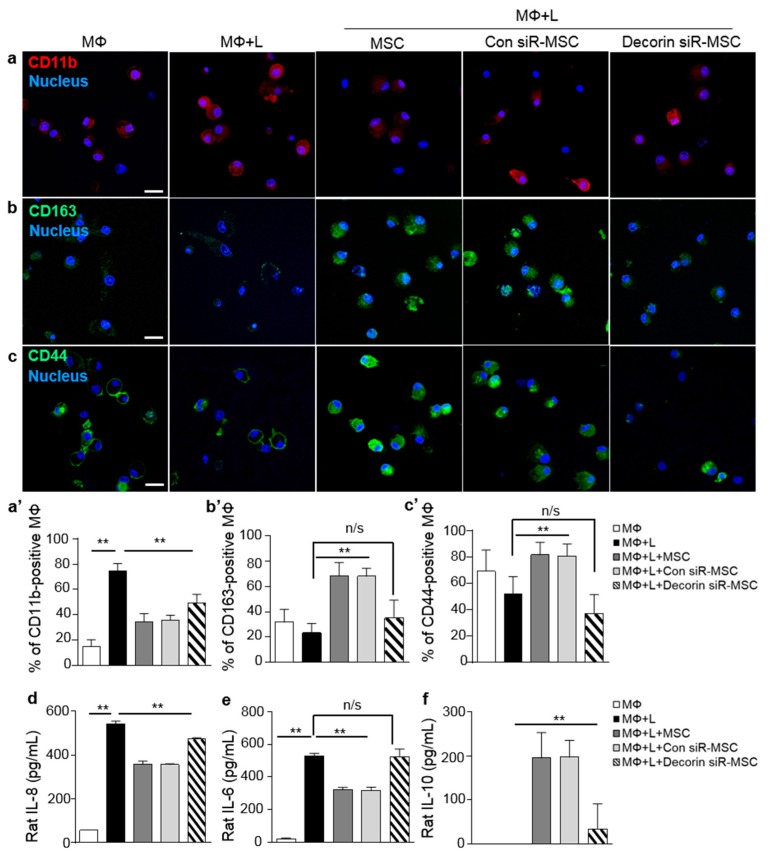
Decorin knockdown in MSCs attenuates macrophage polarization. (**a**–**c**) Decorin siRNA-treated MSCs were co-cultured with LPS-induced macrophages and macrophage polarization was assessed. (**a’**–**c’**) Immunofluorescent staining with CD11b, CD163, and CD44 were analyzed according to the percentage of positively stained cells. Red (CD11b) and green (CD163 and CD44) staining indicate positive cells. Nuclei were stained with Hoechst 33342. (**d**) IL-8, (**e**) IL-6, and (**f**) IL-10 levels in co-cultures of macrophages and MSCs were analyzed by ELISA. Data are presented as mean ± SD, *n* = 5 per group. Scale bar = 20 μm. ** *p* < 0.01. n/s, not significant; MΦ, macrophage; L, LPS.

**Figure 4 ijms-20-04815-f004:**
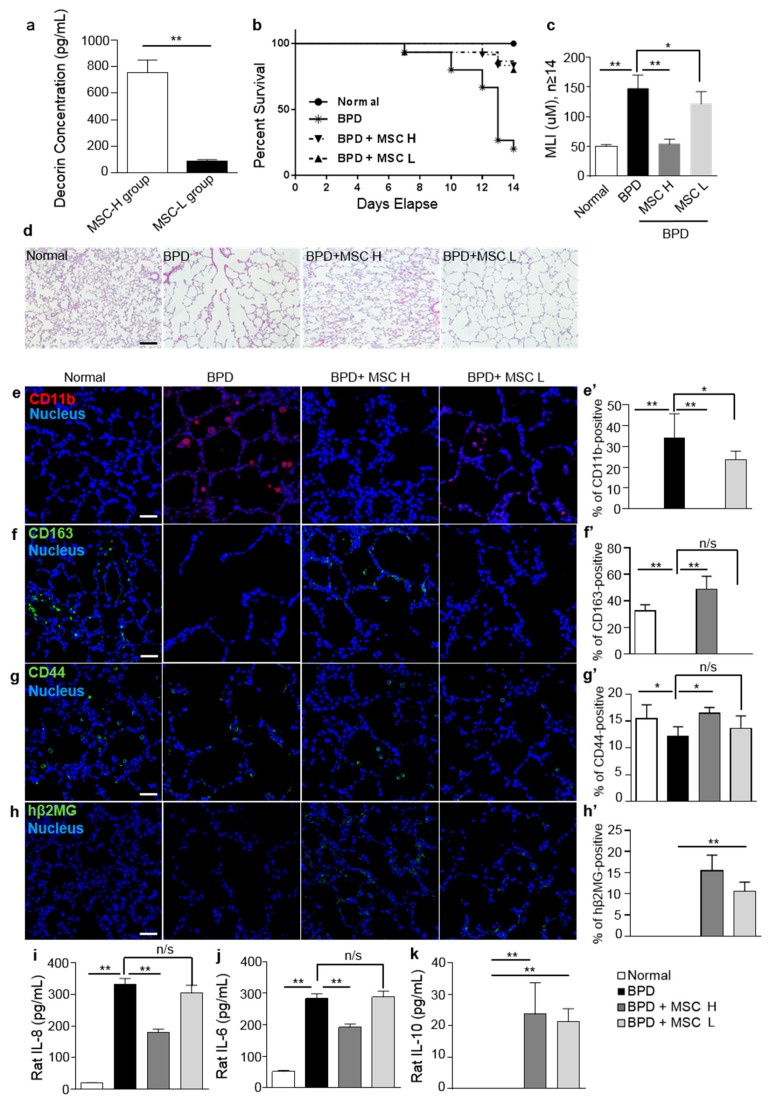
MSCs promote tissue repair in a hyperoxic lung injury rat model. The normal group was raised under normoxic conditions, whereas the experimental groups (bronchopulmonary dysplasia (BPD), BPD + MSC H, or MSC L) were raised under hyperoxia conditions (90% oxygen) from 10 h after birth to postnatal day (P14). On day 5, 1 × 10^5^ MSCs were injected intratracheally. Rat pups were sacrificed and lung tissues and bronchoalveolar lavage fluid (BALF) samples were collected on P14. (**a**) MSCs were divided into two groups (MSC-H and MSC-L) according to their decorin expression levels. Decorin secretion by MSCs was quantified by secretome analysis and ELISA. Supernatants from co-cultures of macrophages and MSCs showed different levels of decorin. (**b**) MSC H lot was selected from MSC-H groups and MSC L lot was selected from MSC-L groups. Kaplan–Meier survival curve at birth and P14. (**c**) Morphometric analysis of lung tissues was performed to determine the degree of alveolarization by measuring the mean linear intercept (MLI). (**d**) Lung tissue samples were sectioned and stained with hematoxylin and eosin (H&E). (**e**–**g**) CD11b, CD163, and CD44 were stained in rat lung tissues. (**h**) Immunofluorescence staining of human β2 microglobulin (hβ2MG) was used to visualize the engrafted MSCs. (**e’**–**h’**) Positively stained cells were analyzed and presented as percentages. Red (CD11b) and green (CD163, CD44, and hβ2MG) staining indicate positive cells. Nuclei were stained with Hoechst 33342. (**i**–**k**) Lung bronchoalveolar lavage fluid (BALF) samples were collected to analyze the pro-inflammatory cytokines, IL-8 and IL-6 and the anti-inflammatory cytokine, IL-10, by ELISA. Data are presented as mean ± SD for *n* = 11 (normal) or 15 (BPD, BPD + MSC H, and MSC L) per group. Scale bar = 40 μm. ** *p* < 0.01, * *p* < 0.05. n/s, not significant; BPD, bronchopulmonary dysplasia.

**Figure 5 ijms-20-04815-f005:**
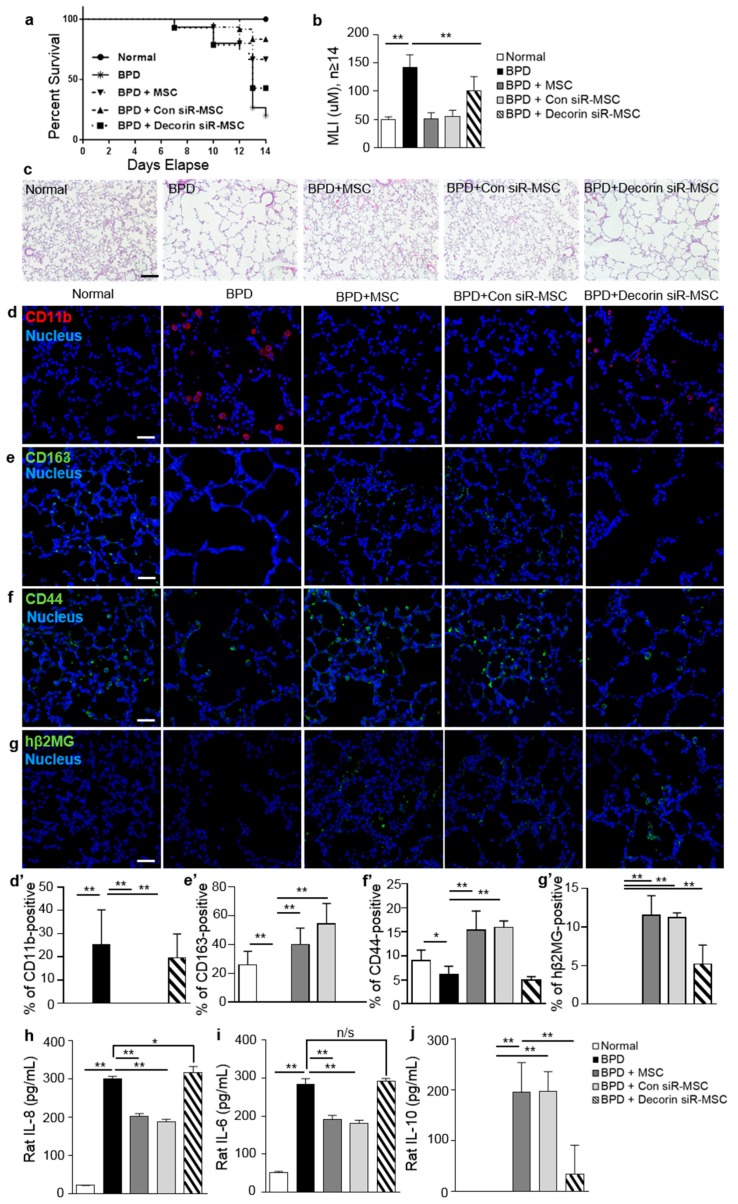
Decorin knockdown suppressed the therapeutic effects of MSCs by down-regulating macrophage polarization. Decorin-silenced MSCs were intratracheally injected in BPD rats. (**a**) Decorin siRNA-transfected MSCs were intratracheally injected on day 5. Daily survival rates during 14 d from birth are presented as Kaplan–Meier survival curves. (**b**) Lung tissues on P14 were analyzed by histological and morphogenic comparison with the control group (normal). The mean linear intercept (MLI) values were determined by the degree of alveolarization. (**c**) Lung tissue samples were sectioned and stained with H&E. Data are presented as mean ± SD for *n* = 11 (normal), 15 (BPD, BPD with MSC), or 14 (co-cultured with MSCs or scrambled siRNA-transfected MSCs (Con siR-MSC), Decorin siR-MSC) per group. ** *p* < 0.01, **p* < 0.05. n/s, not significant; BPD, bronchopulmonary dysplasia. (**d,d’**–**f,f’**) Immunofluorescence analysis was performed with CD11b, CD163, and CD44 antibodies in lung tissues obtained from rats with hyperoxic lung injury. (**g**,**g’**) Transplanted MSCs were detected by hβ2MG (green) staining of rat lung tissues. Blue staining represents nuclei stained with Hoechst 33342. Red (CD11b) and green (CD163 and CD44) staining indicate positive cells. (**h**–**j**) The levels of cytokines IL-8, IL-6, and IL-10, were analyzed by ELISA on P14. Data are presented as mean ± SD, *n* = 5 per group. Scale bar = 40 μm. ** *p* < 0.01, * *p* < 0.05. n/s, not significant; MΦ, macrophage; L, LPS.

**Table 1 ijms-20-04815-t001:** Sequences of primers used for indicated target genes.

Target Gene	Primer Sequence (5′-3′)
Scramble siRNA	UGGUUUACAUGUCGACUAA
	UGGUUUACAUGUUGUGUGA
	UGGUUUACAUGUUUUCUGA
	UGGUUUACAUGUUUUCCUA
Human decorin siRNA	AGAUGAGGCUUCUGGGAUA
	GCUGGACCGUUUCAACAGA
	UCAAUGCCAUCUUCGAGUG
	CGACUUCGAGCCCUCCCUA
Rat CD44 siRNA	ACUAAGAGUGGUCGAAGAA
	GCACAGCAGCAGAUCGAUU
	CAUAGAAGGACACGUGGUA
	GAUACAGGCUCCAGUCAUA

**Table 2 ijms-20-04815-t002:** Experimental design in vivo.

Experimental Design	Group	Rat Numbers	Transplanted MSC Numbers	Volume (μL) of Saline
1	Normal	11	-	50
	BPD	15	-	50
	BPD + MSC H	15	1 × 10^5^	50
	BPD + MSC L	15	1 × 10^5^	50
2	Normal	11	-	50
	BPD	15	-	50
	BPD + MSC	15	1 × 10^5^	50
	BPD + Con siR-MSC	14	1 × 10^5^	50
	BPD + Decorin siR-MSC	14	1 × 10^5^	50

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
