# Peer review of "Decorin Secreted by Human Umbilical Cord Blood-Derived Mesenchymal Stem Cells Induces Macrophage Polarization via CD44 to Repair Hyperoxic Lung Injury"

_ijms, 2019, doi:10.3390/ijms20194815_

Round 1

Reviewer 1 Report

Dear Authors,

The manuscript entitled "Decorin Secreted by Human Umbilical Cord Blood-Derived Mesenchymal Stem Cells Induces Macrophage Polarization via CD44 to Repair Hyperoxic Lung Injury" was very interesting. 

The authors exhibited the potential of MSCs to swift the macrophages from M1 to M2. Also, showed their beneficial effect of the MSCs application to hyperoxic lung injury.

The results are quite significant and may represent in future a potential use in humans. However, major revisions are required before the manuscript will be further processed.

Specifically

1)    2.Results Lines 97-99. The authors mentioned that co-culture of MSCs with macrophages result to reduction of CD11b expression. In figure 1, this is not clear. My proposal to the authors is to use lower magnification in the images in order to demonstrate better their results.

2)   2.Results Lines 100-111. In Figure 1, it is better to number each image in order to be easier for the readers to understand. Also, in diagramms b and c each group should be added in the x axis as in diagram d.

3)    2.Results Lines 117- 119. The authors mentioned that used MSCs that were prepared in different ways. Please mention in more detail the different preparation methods of MSCs. In addition, in figure 2, please add the corresponding numbers in the images.

4)  2.Results Lines 194 - 200. The authors mentioned that they used 10 different lots of MSCs and then they divided into two groups based on decorin levels. What was the origin of these MSCs and how the decorin level was quantified in order to distinguis between high an low expression rates of decorin groups.

5) 2.Results Lines 230 - 231. The results showed that MSCs with lower decorin expression (MSCs -H) presented higher survival rates than MSCs with higher decorin expression. Is this correct?

6)  3. Discussion lines 342 - 343. Please remove the sentence "Decorin and biglycan...gene duplication [54,55].

7) 3. Discussion lines 408 -409. Authors mentioned that the therapeutic effects towards hypeoxia induced lung injuries is mediated by anti inflammatory effects rather than regenerative capacity. However, the authors did not perform experiments regarding the growth factor quantification that could play significant role in the MSCs mediated regeneration. In order to assume that the authors must quantify these  growth factors with known regenerative potential such as TGF-β, EGF, VEGF.

8) 3. Discussion. Please include and discuss the publication of Xaus et al. (Blood 2001, 98:2124 - 2133)

8) 4. Materials and Methods lines 417 - 426.  The authors used for their experiments MSCs derived from human umbilcal cord blood. Umbilical cord blood is not a rich source of MSCs. In addition, extended cultivation time is needed for their expansion. Why the authors used this MSC origin instead of bone marrow, or adipose tissue. Did the authors believe that these MSCs are characterized by better immunoregulatory properties when compared to other sources?

9) 4. Materials and Methods lines 417 - 426  In addition, the MSCs were in passage 6 which is high. It is widely known that erasing the passage of MSCs could alter significantly their properties. By using MSCs from other sources like bone marrow or adipose tissue, this maybe could be avoided. This is just a suggestion. In addition Figure S1 does not show the MSCs characteristics. This is shown in Figure S3. Please perform the correction.

10) 4. Materials and Methods. Lines 464 - 480. The trilineage differentiation ability and the flowcytometric analysis were performed in the same passage as the cells used in experimental procedures?

11) In Figure S3 - Image A where the flow cytometric analysis is shown, the expression is quite low. This is how unlike for MSCs derived from human umbilical cord blood, which is high in hematopoietic stem cells. Please check again your results and also perform flow cytometric analysis for CD45 and CD19 as has been defined by Dominic et al. (Cytotherapy -2006 Vol. 8, No. 4, 315 317).

Overall the results are well presented. But in order to conclude that decorin alone is responsible for the polarization of macrophages towards M2, it is reasonable to include in the experimental procedures the use of recombinant decorin as control group. MSCs secrete a wide variety of growth factors which my play a role in this procedure. Furrhermore, it is mentioned in the conclusion that the protective effects of MSCs are atributed through their anti inflammatory effects rather than their regeneration properties. If you decide to include the use of recombinant decorin maybe this hypothesis to be confirmed.

Author Response

Thank you.

Reviewer 2 Report

The manuscript addresses a very important issue for the human health and the experiment are simple but effective in trying to deepen the opportunity of using mesenchymal cell for treatment of bronchopulmonary dysplasia.

Macrophages silenced for CD44: are they silenced before or after LPS treatment? Maybe it has the same effect but specify it in the text.

The figures are quite big and not immediately clear because of the absence of lettering the plots of images quantification. So, please, add letters also to those (It should be better separate images and their quantifications from the other results (IL amounts), but I gather that will increase too much the numbers of figures)

In supplement figure S1 it is not clear whether the authors are showing protein or RNA fold change

Line 165:  can the authors add a figure or table (should be also in supplement) with the secretome data about the difference in the one after macrophages co-culture? It is not important if decorin is not the only one changing due to reference [37].

Line 230: do the 20% are control rats? Not clear if they are the total and the relationship with the 80/75 % of MSC rats in the text, while it is clear in the figure

The use of MSC expressing different amount od decorin is not well related to the in vitro experiment: in line 195 is reported “MSCs under inflammatory conditions” for the assessment of decorin quantity amount; moreover, those cells were also used for in vivo experiments summarized in figure 4. What does it means that “inflammatory condition”? Were the cells from donors subjected to co-culture with macrophages? or cultured with conditioned media or with LPS before use?

Please, explain line 252 sentence:” The success of MSC injected was confirmed by staining for hβ2MG (Figure 4e)”

How do the authors explain the difference in the survival results of groups MSC-L (75%) and siDCN-MSC (40%) did they measure the levels of decorin; is similarly different?

Tn the author should underline that “These data indicated that the paracrine actions of decorin are responsible for macrophage polarization, via an increase in CD44 expression”? as reported in line 284 by linking it to the result reported in figure 3.

The main point that make the manuscript not in the high rates of interest and significance is the lack of data about the mechanism by which the MSC increase their decorin expression and also decorin act as an inducer of CD44 in macrophages. Nevertheless, this paper can be of importance for medical treatment and for assessing the better pool of cells able to have the maximum results in human health.  

Author Response

Thank you.

Round 2

Reviewer 1 Report

Dear Authors,

The revisions that you were performed were exactly to the point. I am impressed by the addition of the growth factors quantification experiments and the overall presenting quality of the manuscript.

In my opinion by performing these changes the manuscript should be forward to the next stage which is the publication

Yours sincerely.